# Pathways to diagnosis of a second primary cancer: protocol for a mixed-methods systematic review

Lovney Kanguru,[1] Annemieke Bikker,[1] Debbie Cavers,[1] Karen Barnett,[1] David H Brewster,[1,2] David Weller,[1] Christine Campbell[1]

[1]Usher Institute of Population Health Sciences and Informatics, University of Edinburgh, Edinburgh, UK
[2]Scottish Cancer Registry, Information Services Division, NHS National Services Scotland, Edinburgh, UK

**Correspondence to**
Dr Lovney Kanguru;
Lovney.Kanguru@ed.ac.uk

## ABSTRACT

**Introduction** As cancer survivors continue to live longer, the incidence of second primary cancers (SPCs) will also rise. Relatively little is understood about the diagnostic pathway for SPCs, how people appraise, interpret symptoms and seek help for a second different cancer and the experiences (including challenges) of healthcare providers relating to SPCs. This study aims to systematically appraise and synthesise the literature on the pathways to diagnosis of an SPC and the associated patient and healthcare provider experiences.

**Methods** The approach taken includes systematic searches of published and unpublished literature without any date or language restrictions. MEDLINE, Embase, CAB Abstracts, MEDLINE In-Process and non-indexed citations, PsycINFO, Epub Ahead of Print, In-Process and other non-indexed citations, Ovid MEDLINE Daily, CINAHL, ASSIA, Sociological Abstracts, Web of Science, PROSPERO and grey literature will be searched to identify observational, systematic reviews, mixed methods and qualitative studies of interest. Titles, abstracts and full texts will be screened against the inclusion–exclusion criteria by at least two reviewers independently. Relevant outcomes of interest and study and population characteristics will be extracted. Synthesis will be used guided by the Pathways to Treatment model and the Olesen model of time intervals.

**Ethics and dissemination** Ethical approval is not required. This systematic review will provide a deeper understanding of the complex and heterogeneous diagnostic pathways of SPCs, while identifying common themes across the diagnostic interval, routes to diagnosis and patient and healthcare provider experiences. These findings will help provide a nuanced picture of the diagnostic pathway for SPCs that may inform policy and consistent practice. In particular, approaches to early diagnosis for an SPC; including the timing and reasons behind the decision by the patient to seek care, the challenges faced by healthcare providers, and in the development of future interventions to reduce the delay in patient time-to-presentation.

**PROSPERO registration number** CRD42016051692.

## Strengths and limitations of this study

► This is the first systematic review to appraise and synthesise the literature on the pathways to diagnosis of a second primary cancer (SPC) and the associated patient and healthcare provider experiences.

► This study may help provide a nuanced picture of the diagnostic pathway for SPCs that may inform policy and consistent practice in particular approaches to early diagnosis for an SPC, including the timing and reasons behind the decision by the patient to seek care and the challenges faced by healthcare providers, and in the development of future interventions.

► The review is not restricted to any time period or specific languages.

► We anticipate that the available data from qualitative studies may be limited (as informed by an initial scoping exercise) and as such, metaethnography synthesis may not be possible to do.

► It is unlikely that a meta-analysis will be applicable for the quantitative studies included because the scope of this review does not cover survival outcomes along the diagnostic pathway for SPCs.

primary cancers (SPCs) is also expected to grow.[2] SPC has been defined as a new primary cancer that occurs in a person who has had cancer in the past.[3] Evidence suggests that the incidence of SPCs among cancer survivors ranges between 1% and 17% depending on the index cancer site,[4–13] with SPCs estimated to account for as much as 16%–18% of total cancer incidence[14–16] in western countries.

Fear of a cancer recurring and the possibility of getting a new primary cancer or multiple primaries are often a source of worry and psychosocial distress among cancer survivors.[17–19] These cancer-related health worries have been linked to higher levels of mental illness particularly anxiety and depression in long-term survivors.[17] Furthermore, the fear of and past experiences of cancer have been shown to affect patients' help-seeking

## INTRODUCTION

The number of people surviving cancer is steadily increasing.[1] As cancer survivors continue to live longer, in common with the wider population, the incidence of second

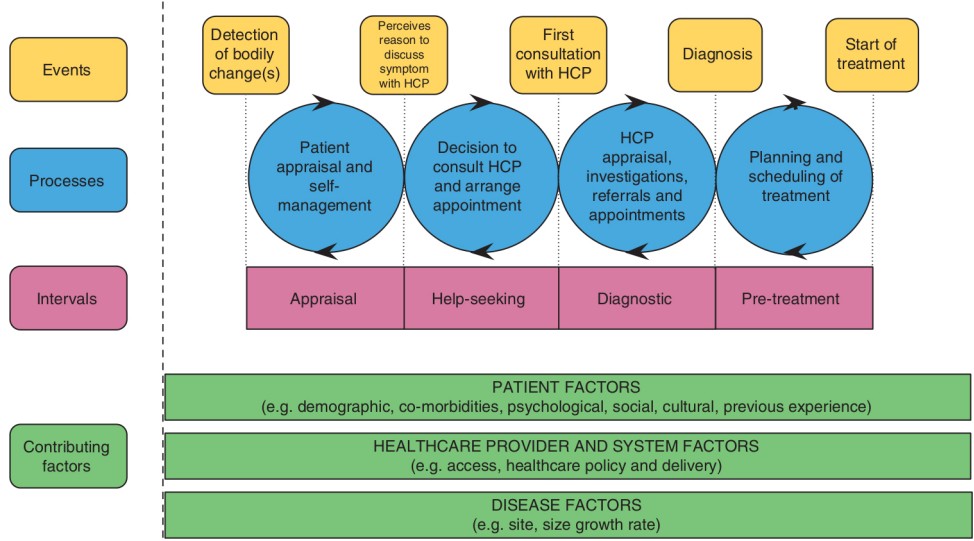

**Figure 1** Pathways to Treatment model[24]. HCP, healthcare professional.

behaviour.[20–23] Symptoms of an SPC may be missed if patients are not aware of their significance.[20] This could be because of patients focusing on the symptoms of the original cancer only or patients' perceptions being influenced by the presence of comorbid conditions.[24–26]

Early detection of cancers is an important element in improving cancer outcomes, as the stage of disease at diagnosis is linked to survival for many cancers.[20 25] While patients' ability to recognise possible warning signs and taking prompt action is a critical early step, evidence suggests that the patient diagnostic pathway is often not straightforward.[27–31] For example, the diagnostic interval may vary depending on the route to diagnosis, which has potential implications for the stage at diagnosis. The pathways to treatment model[24] provides a theoretical framework for the wide range of factors that influence the pathway to diagnosis, those relating to the patient and those linked to the healthcare provider, the health system and the cancer itself (figure 1). There are also key time

points in diagnostic journeys (Olesen *et al*[32] and Weller *et al*[33] as illustrated in figure 2). Both models emphasise the often complex journeys that patients go through from the onset of first symptoms to the start of treatment.

A greater understanding of patient pathways to diagnosis (including diagnostic routes) for an SPC is vital due to the growing incidence, yet comparatively little is known about these pathways. Mapping the evidence on pathways to diagnosis for an SPC and the associated patient and healthcare providers' experiences will be an important starting point that will provide a nuanced picture of the diagnostic pathway for SPCs. These insights may help to inform policy and consistent practice in approaches to early diagnosis for an SPC, including the timing and the decision by the patient to seek care, and in the development of future interventions to reduce the delay in patient time-to-presentation.

This review aims to systematically appraise and synthesise the literature on pathways to diagnosis of an SPC and

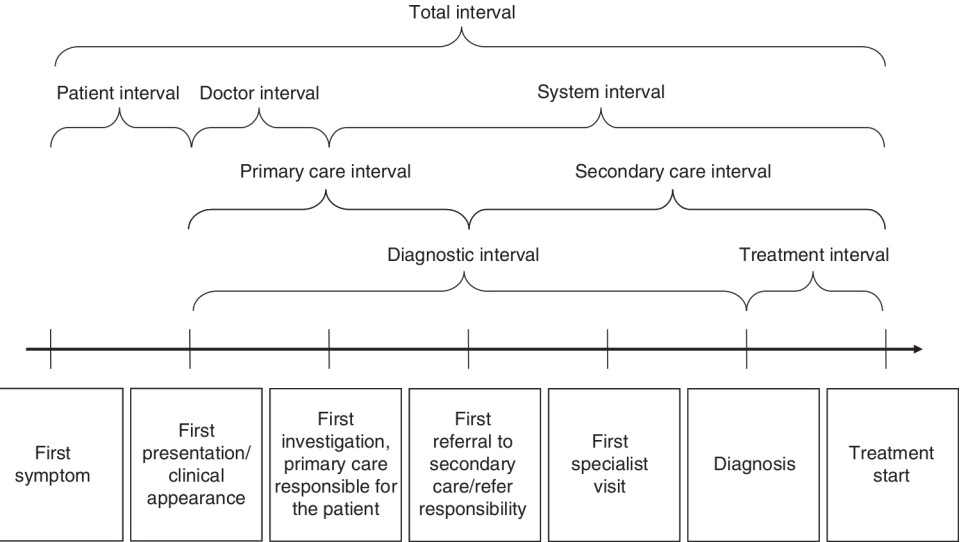

**Figure 2** Model of key time points and intervals (as adapted from Weller *et al*).[33]

**Table 1** Inclusion–exclusion criteria

| | Inclusion | Exclusion |
|---|---|---|
| Pathways to diagnosis | ▶ Any stage of the pathway to diagnosis for an SPC<br>▶ Appraisal, help-seeking, diagnostic and pretreatment intervals<br>▶ Routes to diagnosis | ▶ Pathways to diagnosis for recurrent FPCs or metastatic cancers<br>▶ Studies focusing only on the time between the diagnosis of an FPC and SPC |
| Patients | ▶ SPCs in patients above 18 years of age<br>▶ Patients diagnosed with FPC at 18 years and above | ▶ SPCs in children and adolescents<br>▶ SPCs in adults following a childhood or adolescent FPC |
| Healthcare providers | ▶ Healthcare providers involved in the diagnosis and management of patients with SPC in the diagnostic pathway | |
| Key outcomes of interest | ▶ Pathways to diagnosis for SPCs as per the inclusion criteria above<br>▶ How patients appraise, interpret symptoms and engage with healthcare in the diagnosis of an SPC<br>▶ Healthcare provider experiences, perspectives and challenges faced around the pathways to diagnosis of an SPC | ▶ Patient and healthcare provider experiences related to pathways to diagnosis for recurrent FPCs or metastatic cancers |
| Setting | ▶ Primary, secondary or tertiary care in any country | |
| Study designs | ▶ Cohort, case–control and cross-sectional studies<br>▶ Systematic reviews and mixed methods<br>▶ Qualitative studies such as interviews, focus group discussions and ethnography on the pathways to diagnosis for any SPC and associated patient and healthcare provider experiences | ▶ RCTs<br>▶ Studies using secondary datasets or conference abstracts, conference proceedings and book chapters that are not focused on the diagnostic pathways for an SPC or patient/provider experiences of SPCs<br>▶ Case reports |

FPC, First Primart Cancer; RCT, Randomised Control Trial; SPC, Second Primary Cancer.

associated patient and healthcare provider experiences. Our objectives are:

1. To summarise the literature describing pathways to diagnosis of an SPC, including any descriptions of patient and diagnostic intervals and routes to diagnosis;
2. To summarise the literature describing how patients appraise symptoms and engage with healthcare in the pathway to diagnosis of an SPC;
3. To explore the available evidence on healthcare providers' perspectives around the pathways to diagnosis of an SPC and the challenges they face.

## METHODS

The PRISMA-P (Preferred Reporting Items for Systematic Review and Meta-Analysis Protocols) statement[34] has guided the development and reporting of this systematic review protocol.

### Inclusion and exclusion criteria

Table 1 summarises the inclusion–exclusion criteria. This review will consider studies that report on any stage of the pathways to diagnosis for an SPC, including routes to diagnosis (eg, during routine follow-up or surveillance relating to the first cancer, through engagement with screening, symptomatic presentation in general practice or through emergency presentation). We will also consider studies that report on patient and diagnostic intervals such as symptom appraisal, help-seeking behaviour, diagnostic and pretreatment intervals.[24 28 33] We will follow the definitions of appraisal, help-seeking, diagnostic and pretreatment intervals as defined in the Aarhus statement.[33] Although these definitions are important in this review, we will not exclude studies because of lack of or poor definitions only.

Studies that report on how patients appraise and interpret symptoms, engage with healthcare in the diagnosis of an SPC and on routes to diagnosis, will also be included. Studies for inclusion will be limited to those that examine patients with SPCs only and where patients are 18 years and above at the time of first primary cancer (FPC) diagnosis. We recognise that SPCs in adults following a childhood or adolescent primary cancer is an important area, and there is substantial work that has been done on the late effects of childhood cancers that include SPCs.[35] However, this population group has unique characteristics, thus such studies will not be eligible for inclusion.

We will include studies that report on healthcare providers' experiences including perspectives on the challenges they face around the pathways to diagnosis of an SPC. Exclusions will be made for studies that report

on pathways to diagnosis for recurrent FPCs or metastatic cancers and their associated patient and provider experiences.

We will consider studies from any country conducted in primary, secondary or tertiary setting that are focused on the pathways to diagnosis for an SPC. Furthermore, experiences of healthcare providers around the pathways to diagnosis for an SPC in any setting will be eligible for inclusion.

Eligible study designs include observational studies such as cohort, case–control and cross-sectional studies that report on the pathways to diagnosis for any type of SPC. Systematic reviews, mixed methods and qualitative studies such as interviews, focus group discussions and ethnography reporting on the pathways to diagnosis for any SPC and associated patient and healthcare provider experiences will also be eligible for inclusion. It is unlikely that study designs such as randomised controlled trials (RCTs) will describe the pathways to diagnosis for an SPC and/or patient and provider experiences along that pathway as they are often focused on interventions, thus, such studies will be excluded. Furthermore, conference abstracts, case reports, book chapters and studies that use secondary datasets that are not focused on the diagnostic pathways for an SPC or patient/provider experiences will be excluded. In addition, if these studies do not have complete study methodologies, findings and discussion sections, they will also be excluded from the review.

### Search strategy

We will search published and unpublished literature with no restrictions on the date or language to minimise selection bias. A search strategy of relevant Medical Subject Headings terms was designed, piloted and then further refined (box 1).

To identify relevant published studies, the search strategy will be run initially in MEDLINE through the OVID platform. It will then be adapted, and suitable changes will be made before re-running it again in Embase, CAB Abstracts, MEDLINE In-Process and non-indexed citations, PsycINFO, Epub Ahead of Print, In-Process and other non-indexed citations and Ovid MEDLINE Daily. Other sources such as CINAHL, ASSIA, Sociological Abstracts, Web of Science and PROSEPRO will also be searched, and the search strategy adapted accordingly. References lists of relevant studies identified will also be scrutinised for any potential additional studies.

To identify relevant studies from unpublished literatures, we will search grey literature in Open Grey, ProQuest Dissertations and Theses Global databases for reports and theses.

### Screening and data extraction

All the citations found from the various databases will be imported into EndNote X7. Independently, two reviewers (LK and AB) will screen the titles and abstracts against the inclusion and exclusion criteria. Another two reviewers (CC and DC) will independently screen a

---

**Box 1  Example search strategy for MEDLINE(R)**

1. exp second primary cancer/
2. second primary cancer$.ti,ab.
3. second primary malignan$.ti,ab.
4. second primary tumo$r$.ti,ab.
5. second primary neoplasm$.ti,ab.
6. or/1–5 [SPC TERMS]
7. (patient experience$ or patient know$ or patient perception$ or patient perspective$ or patient view$ or patient interpret$ or patient understand$ or patient aware$ or patient attitude).ti,ab.
8. ((health$care provider$ or health$care personnel or health$care professional$ or health$care practitioner$ or doctor$ or general practitioner$ or GP$ or consultant$) adj2 (experience$ or challenge$ or view$ or understanding$)).ti,ab.
9. or/7–8 [PATIENT & PROVIDER TERMS]
10. pathway$ to diagnos$.ti,ab.
11. diagnos$ journey.ti,ab.
12. diagnos$ pathway$.ti.
13. diagnos$ pathway$.ti,ab.
14. (diagnos$ adj2 pathway$).ti,ab.
15. route$ to diagnos$.ti,ab.
16. diagnos$ route$.ti,ab.
17. pathway$ to treatment$.ti,ab.
18. (patient appraisal or help$seeking behavio$r$ or diagnos$ interval$ or pre$treatment interval$).ti,ab.
19. (first symptom or first presentation or clinical appearance or first investigation or first referral or first specialist visit or diagnosis or treatment start).ti,ab.
20. ((patient appraisal and self$ management) or decision to consult health$care provider or health$care provider appraisal, investigations, referrals, appointment$ or planning or scheduling treatment).ti,ab.
21. or/10–20 [PATHWAYS TO DIAGNOSIS TERMS]
22. 9 or 21 [PATIENTS & PROVIDER TERMS AND PATHWAYS TO DIAGNOSIS TERMS)
23. 6 and 22 [CANCER TERMS AND PATIENTS & PROVIDER TERMS AND PATHWAYS TO DIAGNOSIS TERMS COMBINED]
24. limit 23 to humans [LIMITED TO HUMAN ONLY STUDIES]

---

minimum of 10% of these titles and abstracts to check for consistency. Full texts of all potentially relevant articles will be retrieved and again screened by at least two reviewers independently to ascertain whether they have met the inclusion–exclusion criteria. Any disagreements will be resolved through team discussions.

A data extraction form will be designed and piloted to ensure that it captures all the relevant data. The included studies will be divided between reviewers, where one reviewer will carry out the data extraction of the first half and the second reviewer will carry out the data extraction of the second half. Both reviewers will do this independently and then countercheck each other's data extraction.

The data extraction form will include key characteristics of the studies such as author, publication year, period of the study (by year(s)), data source, country of study, study design and setting (ie, primary, secondary or tertiary care) and type of healthcare system including first point of contact with the healthcare provider. In

**Table 2** Quality assessment tools

| Tools | Study designs |
|---|---|
| CASP checklist | Systematic review<br>Case–control<br>Cohorts<br>Qualitative (eg, interviews, focus group discussions and ethnography) |
| McMaster Critical Review Tool and STROBE | Cross-sectional<br>Before and after |
| MMAT | Mixed methods |
| Aarhus checklist | All included studies that report and measure intervals of patient cancer journey |

CASP, Critical Appraisal Skills Programme; MMAT, Mixed Methods Appraisal Tool; STROBE, STrengthening the Reporting of Observational Studies in Epidemiology.

addition, characteristics of the study population such age, gender, ethnicity and cancer types will be included which will provide the context for explanations and interpretations. Any information on the pathways to diagnosis for an SPC will be extracted. This includes routes to diagnosis, the onset of first symptoms, first presentation or first contact with the healthcare provider, first investigation of cancer-related symptoms, referral to hospital, first specialist visit, diagnosis and start of treatment. Furthermore, information on appraisal, help-seeking, diagnostic and pretreatment intervals will also be extracted. Any other relevant patient and provider experiences (including challenges) for SPCs along the diagnostic pathway will also be extracted.

## Quality assessment

Quality assessment will be carried out in parallel to data extraction. The quality assessment tool will vary depending on the study design of the included article. There is currently no consensus as to which instrument is 'best' for assessing observational studies, especially of non-intervention studies. Therefore, we plan to adapt several published tools as shown in table 2. We will use the Critical Appraisal Skills Programme (CASP) tool for systematic reviews, case–control and cohort studies. The CASP tool for qualitative studies will also be used. Within the literature on the synthesis of qualitative studies, it has been argued that appraisal tools focus more on the methodological rather than the conceptual strengths.[36 37] Therefore, in order to ensure that the authors' theoretical insights are captured, qualitative studies will not be excluded because of the quality of the methods alone.

The McMaster Critical Review tool will be adapted for cross-sectional and before after studies guided by the STROBE statement.[38–40] The Mixed Methods Appraisal Tool developed by the National Collaborating Centre for Methods and Tools at McMaster University will be used for mixed methods studies.[41]

We will also apply the Aarhus checklist as a framework for describing the rigour of papers to studies that describe and measure intervals on the pathways to cancer diagnosis for an SPC. The Aarhus checklist was developed as a resource for early diagnosis research that describes and measures intervals in patient cancer journeys.[33]

## Data synthesis

We are uncertain of the profile of study types that we will uncover, although we anticipate most will be descriptive studies. As such, data synthesis will depend on the type of papers identified. Tables and diagrams or charts will be used to summarise the findings as appropriate. Depending on the included papers, synthesis will consider different categorisations of findings to explore. For example, varying combinations of FPC and SPCs, patterns over time and different healthcare delivery systems.

For the qualitative studies, we anticipate that the available data will be limited (as informed by an initial scoping exercise) and as such, metaethnography analysis will not be appropriate. Therefore, a narrative framework will be used to summarise and describe the available evidence of pathways to diagnosis of an SPC.[42] Similarly, narrative synthesis will be used to summarise findings from quantitative studies. It is unlikely that a meta-analysis will be applicable because the scope of this review is not intended to cover survival outcomes along the diagnostic pathway for SPCs. Where information on routes to detection is available, these will be analysed separately.

Increasingly, narrative synthesis in systematic reviews has been used as a way of collating evidence from research that has been conducted by a range of methods.[43] Although it has been criticised for its lack of transparency, we plan to be explicit in how we identify key concepts and themes across individual studies. The two models of pathways to cancer diagnosis mentioned earlier, the Pathways to Treatment model[24] and the Olesen model of time intervals[32] as adapted by Weller *et al*[33] will guide the synthesis.

We will first start by developing a preliminary synthesis and exploring the relationships within and across studies. NVIVO 10 will facilitate the process of comparing and contrasting information across the included studies where possible, which will allow for indexing and coding of themes to be done systematically. The focus of the synthesis will be to map out a comparison of the included studies against the model of Pathways to Treatment and the model of time intervals. It is anticipated that this comparison will lead to the identification of key findings of similarities and differences within and across the included studies.

**Acknowledgements** The authors like to express their sincere thanks and gratitude to Marshall Dozier for her advice on the searches and EndNote.

**Contributors** The systematic review was conceived by DC, KB, CC and DW. LK and AB designed the systematic review protocol with inputs from DC, CC, DW, DHB and KB. LK drafted the manuscript with subsequent input from other authors. CC is the guarantor and the principal investigator of this review.

**Funding** The systematic review is part of a larger study looking at understanding pathways to diagnosis, patient and providers' experiences and survival outcomes for second primary cancers, funded by the Cancer Research UK (reference code C12357/A21326).

**Competing interests** None declared.

**Ethics approval** The University of Edinburgh Centre for Population Health Sciences Ethical Review Committee reviewed and approved this study.

**Provenance and peer review** Not commissioned; externally peer reviewed.

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
