## [Reviewer comments · BMJ Open]

ARTICLE DETAILS

TITLE (PROVISIONAL)	Pathways to diagnosis of a second primary cancer: protocol for a mixed methods systematic review
AUTHORS	Kanguru, Lovney; Bikker, Annemieke; Cavers, Debbie; Barnett, Karen; Brewster, David; Weller, David; Campbell, Christine

VERSION 1 - REVIEW

REVIEWER	Danny Youlden Cancer Council Queensland, Australia
REVIEW RETURNED	31-May-2017

GENERAL COMMENTS	Thank you for the invitation to review this paper, which describes the protocol for a systematic literature review on the pathways to diagnosis for patients with a second primary cancer. The methods described are quite comprehensive, and I look forward to reading the completed review. My comments for consideration by the authors involve only minor revisions: Major Revisions Nil Minor Revisions 1. Page 7 line 12 – Studies that report on any type of SPC will be considered. There are obviously many possible combinations of FPC and SPC. It would be helpful if the authors could expand on how this will be handled in the review, as it seems likely that the pathway to diagnosis may vary depending on the types of cancer involved. 2. Page 7 line 38 – There are no restrictions on the date or language of the included literature. How will potential changes over time be identified and handled? Are there likely to be any barriers regarding misinterpretation of different languages? Discretionary Revisions 3. Page 6 line 13 – Table 2 is mentioned in the text before Table 1 (page 7 line 43).
--

REVIEWER	Jong Hyock Park Chungbuk National University, South Korea
REVIEW RETURNED	25-Jun-2017

GENERAL COMMENTS	I agree with this research purpose and research methods are well
--

	structured. However, the authors should consider that the information on pathways to diagnosis for a SPC such as routes to diagnosis, first contact with the healthcare provider, first specialist visit or something largely depends on the national primary care system and the healthcare delivery system in each country. Therefore, countries should be categorized according to the major factors that may affect the utilization of medical services, the results of the countries in each category should be collected. Minor error: Reference # 11: The title of the paper is mislabeled.
--	--

VERSION 1 – AUTHOR RESPONSE

Reviewer: 1

Reviewer Name: Danny Youlden

Institution and Country: Cancer Council Queensland, Australia Please state any competing interests or state 'None declared': None declared

Please leave your comments for the authors below Thank you for the invitation to review this paper, which describes the protocol for a systematic literature review on the pathways to diagnosis for patients with a second primary cancer. The methods described are quite comprehensive, and I look forward to reading the completed review.

My comments for consideration by the authors involve only minor revisions:

Major Revisions

Nil

Minor Revisions

1. Page 7 line 12 – Studies that report on any type of SPC will be considered. There are obviously many possible combinations of FPC and SPC. It would be helpful if the authors could expand on how this will be handled in the review, as it seems likely that the pathway to diagnosis may vary depending on the types of cancer involved.

Authors' response: Thank you for this comment. We agree that this is potentially a significant area. The aim of this review is to map the evidence on pathways to diagnosis for a SPC and the associated patient and health-care providers' experiences. This includes exploring in what ways the pathway to diagnosis may vary depending on the types of FPC and SPC. If the review identifies variations in the pathways then we will report those in the results. We have added a sentence to the data synthesis section on page 10 to make this clear.

2. Page 7 line 38 – There are no restrictions on the date or language of the included literature. How will potential changes over time be identified and handled?

Authors' response: We agree that changes over time could have an effect on the pathways to diagnosis. We have added a few words to the screening and data extraction section on Page 8 to reflect how potential changes over time will be identified by extracting information on the period of the study (by year). We have also added a sentence on page 10 to show how this will be handled depending on the studies included.

Are there likely to be any barriers regarding misinterpretation of different languages?

Authors' response: Thank you for raising the issue of potential misinterpretation. We do not anticipate any problems with this, although we have noticed that there is a great deal of variation in the terms and definitions used for SPCs. We will be cautious of this when conducting the review and will report of any potential issues noted in the completed review.

Discretionary Revisions

3. Page 6 line 13 – Table 2 is mentioned in the text before Table 1 (page 7 line 43).

Authors' response: Thank you for bringing this to our attention. We have reordered and renumbered the tables (pages 19, 20), and adjusted this in the text in the inclusion and exclusion criteria section in page 6 and the search strategy section in page 7

Reviewer: 2

Reviewer Name: Jong Hyock Park

Institution and Country: Chungbuk National University, South Korea Please state any competing interests or state 'None declared': None

Please leave your comments for the authors below I agree with this research purpose and research methods are well structured. However, the authors should consider that the information on pathways to diagnosis for a SPC such as routes to diagnosis, first contact with the healthcare provider, first specialist visit or something largely depends on the national primary care system and the healthcare delivery system in each country. Therefore, countries should be categorized according to the major factors that may affect the utilization of medical services, the results of the countries in each category should be collected.

Authors' response: Thank you for these points.

Regarding consideration of information on routes to diagnosis, the review will be looking at this, which has been highlighted on page 6 under inclusion and exclusion criteria.

Regarding the point on 'first contact with the healthcare provider, first specialist visit or something largely depends on the national primary care system and the healthcare delivery system in each country', we have added a sentence to reflect this on page 8 under screening and data extraction, and on page 10 under data synthesis.

Regarding the point on categorisation of countries according to the major factors that affect the utilisation of medical services, we agree that this an important element. We will consider aspects of different healthcare systems in our synthesis that may influence (overall) pathways to diagnosis.

Minor error:

Reference # 11: The title of the paper is mislabeled.

Authors' response: Thank you for pointing this out. It appears that the title of the original article has been published as such (with an error) as shown on this pubmed link: <https://www.ncbi.nlm.nih.gov/pubmed/25448459>. We are happy to take advice from the BMJ Open Editorial office on the matter.

VERSION 2 – REVIEW

REVIEWER	Jong Hyock Park Chungbuk National University, South Korea
REVIEW RETURNED	11-Aug-2017

GENERAL COMMENTS	The author revised the paper to reflect the comments of the reviewers. I have no other opinions.
--